# CURATING HIGH QUALITY PRETRAINING DATA FOR LANGUAGE MODELS VIA COMPRESSION RATIOS

## ABSTRACT

The quality of pretraining data determines the capabilities of language models, yet identifying high-quality data among billions of web documents remains computationally prohibitive. We introduce `Compel`, a simple and scalable data processing step that isolates high-quality text using lightweight, compression-based signals. Our key insight is that the compression ratio of text serves as a robust, model-free proxy for information density: low compression ratios typically reflect repetitive or boilerplate content, whereas high ratios may indicate noisy or unnatural text (e.g., HTML spam or phone numbers). `Compel` improves dataset quality by retaining only those documents whose compression ratios fall within a chosen range, determined empirically from high-quality reference datasets, without relying on additional model training or heuristic classifiers. `Compel` improves benchmark performance by around 0.5–1.1% across leading open web-scale datasets - DCLM, FineWeb, and FineWeb-EDU - all while requiring only a fraction of the computational resources of traditional filtering methods. These results show that compression-based filtering is a practical, compute-efficient complement to prevailing quality controls, capable of boosting pretraining data quality.

## 1 INTRODUCTION

Recent advances in language models have been driven by pretraining on massive web-scale corpora (Brown et al., 2020; Du et al., 2022; Hoffmann et al., 2022; Raffel et al., 2020a; Xie et al., 2023). However, the sheer size of these datasets—often comprising trillions of tokens—makes efficient data selection both crucial and challenging (Albalak et al., 2024).

Carefully curated pretraining data improves downstream performance, as shown by corpora like FineWeb-EDU (Penedo et al., 2024) and DCLM (Li et al., 2024). These datasets rely on multi-stage filtering pipelines—combining language identification, duplication and toxicity checks, and scoring heuristics based on LLMs or perplexity (Gehman et al., 2020; Lee et al., 2022; Marion et al., 2023; Albalak et al., 2024). While these pipelines are effective, the notion of "high quality" remains under-defined, and existing filters often reflect implicit or task-specific assumptions (Longpre et al., 2023). As pretraining datasets grow to tens of trillions of tokens, we need fast and interpretable signals to help identify high-quality data at scale.

This raises a core question: *How can we identify high quality data cheaply and quickly?*

There are likely many promising answers to this question. In this work, we introduce a novel approach that is complementary to existing methods and easily integrates into current filtering pipelines. We propose `Compel`, a scalable and lightweight filtering method that uses a document's compression ratio as a proxy for information density. In information theory, data compression involves encoding information using fewer bits than the original representation, effectively reducing redundancy without losing essential content (Shannon, 1948). Our key insight is that data compression provides a surprisingly effective signal for identifying high-quality text. Intuitively, well-edited, information-rich documents contain meaningful content that is neither overly repetitive nor excessively noisy. These documents tend to compress moderately. In contrast, boilerplate or repetitive content compress too well (e.g., thousands of identical HTML tags), while noisy or synthetic data compress poorly due to its lack of structure. By discarding documents with extremely low or high compression ratios, `Compel` improves the overall information density of a corpus. We calibrate filtering thresholds using high-

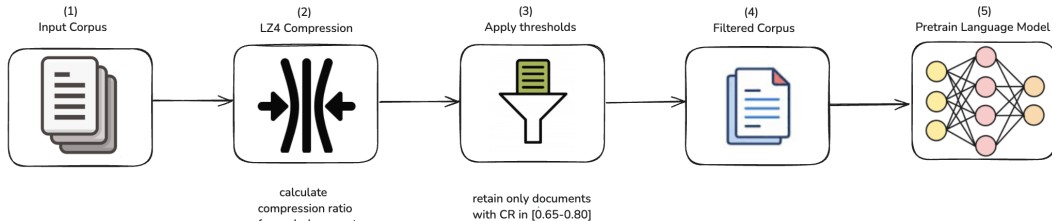

Figure 1: **Figure 1: `Compel` filters low-quality documents using compression ratio.** (1) Start with a web-scale input corpus. (2) Compute the compression ratio (CR) of each document using LZ4 compression. (3) Discard documents with CR outside a calibrated quality band (e.g., $[0.65, 0.80]$). (4) Retain the filtered corpus for pretraining. (5) Pretrain Language Models with improved dataset. `Compel` improves data quality using a fast, model-free signal that requires no training, inference, or supervision.

quality reference datasets and show that this lightweight filter can further enhance performance—even when applied on top of state-of-the-art heuristic or model-based filtering.

We validate our method on three leading pretraining corpora—FineWeb, FineWeb-EDU, and DCLM—each of which employs a distinct filtering strategy. Across 13 downstream tasks, models trained on `Compel`-filtered data consistently outperform those trained on the original corpora. These improvements are robust and achieved at negligible computational cost. While the resulting accuracy gains are relatively modest (+0.5–1.1 points), recent studies have shown that even small consistent gains from lightweight filters can be substantial at scale, offering considerable efficiency benefits when applied to trillion-token datasets (Raffel et al., 2020a; Wenzek et al., 2020; et al., 2024a).

## 2 RELATED WORK

Data selection pipelines for pretraining large models typically involve a sequence of filtering stages to manage the scale and heterogeneity of web-derived corpora. Common goals include removing undesirable content-such as duplicated boilerplate, SEO spam, or template-heavy pages-and maximizing overall data quality.

**Language filtering.** Language identification is usually performed using lightweight classifiers such as fastText or langdetect (Raffel et al., 2020b; Soldaini et al., 2024; Conneau & Lample, 2019; Xue et al., 2021; Laurençon et al., 2022; Grave et al., 2018). These methods are efficient but require careful threshold tuning to avoid excessive false positives or negatives.

**Heuristic filtering.** Surface-level heuristics such as document length or the presence of blacklisted terms are commonly used (Rae et al., 2021; Raffel et al., 2020b; Aghajanyan et al., 2022). These are computationally cheap but often fail to capture more nuanced indicators of quality.

**Quality filtering.** Retaining educational, human-written text is a key objective in many pipelines. Classifier-based approaches typically train fastText models or hashed-feature classifiers on curated datasets (Du et al., 2022; Longpre et al., 2023), while perplexity-based methods score documents using language models trained on high-quality corpora (Wenzek et al., 2020; Wettig et al., 2024). Despite strong performance, these methods face limitations: (1) reference corpora may encode cultural or demographic preferences, introducing bias that can marginalize certain dialects, domains, or communities (Gururangan et al., 2022; Xu et al., 2021), thereby reinforcing existing inequities in model behavior and (2) the notion of "quality" remains inherently subjective and task-dependent (Longpre et al., 2023). In contrast, `Compel` sidesteps these challenges by avoiding reliance on human-annotated or culturally-specific reference corpora, instead using a purely statistical signal (compression ratio) that generalizes across domains.

**LM-based quality filtering** Recent approaches have employed language models for data refinement. For instance, (et al., 2024b) introduced Web Rephrase Augmented Pre-training (WRAP), which uses instruction-tuned models to paraphrase web documents into styles like Wikipedia or question-answer formats, enhancing data quality and training efficiency. Similarly, (Su et al., 2024) presented Nemotron-CC, a refined dataset derived from Common Crawl, combining classifier ensembling,

| **CR** $< 0.65$ | $0.65 \leq$ **CR** $\leq 0.80$ | **CR** $> 0.80$ |
| --- | --- | --- |
| Asus Laptop Virus Removal Asus Repair Service Area: Asus Laptop Virus Removal Toronto, Laptop Repair GTA, Laptop Repair Thornhill, Laptop Repair Richmond Hill, Laptop Repair Markham, Laptop Repair Vaughan, Laptop Repair Toronto, Laptop Repair Scarborough, Laptop Repair Newmarket, Laptop Repair Mississauga... | This project is solving the Asteroid Watchers challenge. AROs are essentially created by combining a telescope with a smartphone. If the telescope has drive motors controllable via a standard // protocol, it is commanded by the SkyWatch app on the phone. Telescopes without motors need an Arduino-driven stepper setup. We plan to publish a... | 根香港法律,不得在程中,向未成年人售或供令人醺醉的酒。 Under the law of Hong Kong, intoxicating liquor must not be sold or supplied to a minor in the course of business. Enjoy FREE DELIVERY for orders HK$1500 (net) or more, otherwise a delivery charge may apply. Hotline: +852 9063 3222 visa visa verified master master secure paypal... |

Figure 2: Beginning characters of representative examples excluded for low compression ratio (left), kept by `Compel` (middle), and excluded for high compression ratio (right). `Compel` retains the informative mid-range while filtering boilerplate or noisy text.

synthetic data rephrasing, and reduced reliance on heuristic filters to balance data quality and quantity. While effective, these methods are computationally intensive, requiring substantial resources for model inference and data generation.

**Compression-based data selection.** The link between data compression and language modeling has long been observed (Shannon, 1951). Recent studies reinforce this connection: LMs can act as powerful compressors (Delétang et al., 2023), and generalization ability may correlate with compression capacity (Huang et al., 2024). (Pandey, 2024) proposes a data-dependent scaling law based on gzip compressibility. Compression has recently emerged as a scalable, embedding-free signal for data selection and quality estimation. (Yin et al., 2024) selects diverse alignment data by scoring examples based on their incremental compressed size, optimizing corpus diversity. ZIP-FIT (Obbad et al., 2024) uses Normalized Compression Distance (NCD) to filter source data for fine-tuning tasks, outperforming neural embeddings in settings such as code and autoformalization.

`Compel` differs from ZIP and ZIP-FIT in both goal and design. ZIP aims to construct diverse, low-redundancy subsets for alignment by selecting examples with low incremental compressed size, relying on early-stage training signals and task-specific assumptions. ZIP-FIT, in contrast, optimizes similarity to a target task distribution using pairwise Normalized Compression Distance. `Compel`, by comparison, is a general-purpose filter designed for broad pretraining use: it discards overly redundant or noisy documents based purely on raw compression ratio. It requires no task-specific supervision, no pairwise comparisons, and no access to model loss signals. Its simplicity and speed make it especially attractive for trillion-token pipelines.

## 3 COMPRESSION RATIO AS A SIGNAL OF DATA QUALITY

We propose compression ratio as a lightweight, model-free proxy for textual data quality. Let $x$ be a document and $c(x)$ its compressed representation using a lossless compression algorithm $c$. We define the *compression ratio* of a string $x$ as:

$$\text{CR}(x) = \frac{\text{num\_bytes}(c(x))}{\text{num\_bytes}(x)}$$

where $\text{num\_bytes}(\cdot)$ denotes the number of bytes. In this work, we use the LZ4 compressor [1], selected for its speed and streaming compatibility. While we would like to try more compressors, and while using different compressors is cheap and fast, pretraining LMs is not. Thus, due to a limited compute budget, we chose to prioritize a single compressor on multiple datasets and multiple language models, over multiple compressors on fewer datasets and LMs because strong evidence for 1 compressor is better than weak evidence for multiple compressors. To the best of our knowledge, our results do not rely on any specifics of LZ4, and so we conjecture that other compressors would work equally well; however, experiments would necessary to test this and we sadly lacked the compute required.

---

[1] `https://lz4.github.io/lz4/`

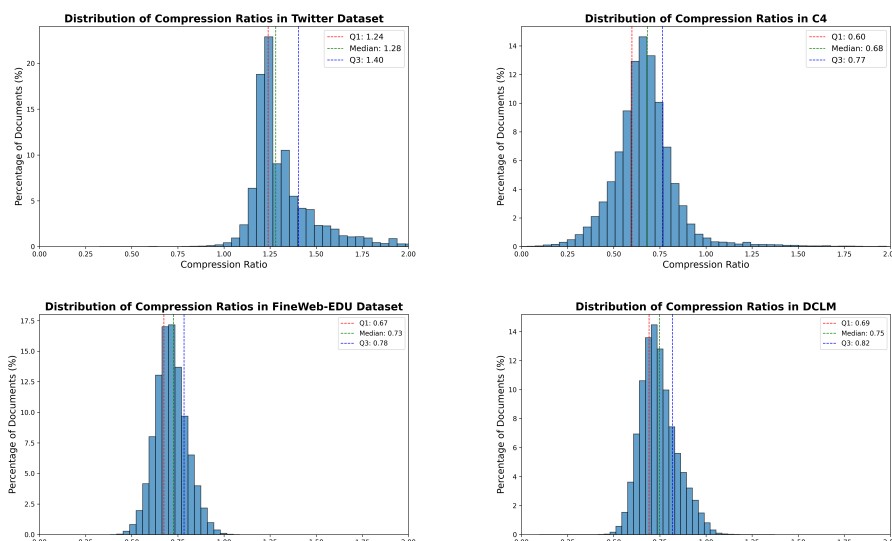

Figure 3: **Compression ratio distributions across datasets.** High-quality datasets like DCLM exhibit tight compression ratio distributions, while noisier corpora like C4 and Twitter show broader or skewed distributions. Each plot represents compression ratio distributions across 1 million samples from the respective dataset, computed using LZ4. **Top left:** C4 (minimally filtered) skews lower with heavy tails. **Top right:** FineWeb-Edu shows a more centralized distribution. **Bottom left:** DCLM (highly curated) peaks sharply near 0.75. **Bottom right:** Twitter (noisy, informal) skews high, reflecting structural noise and short, unstructured content.

Figure 5 shows examples of documents across different compression ratio bands. Manual examination suggests that documents with compression ratios in a Goldilocks zone seem to correspond with higher quality text. Low compression ratio documents frequently exhibit repetitive, templated text or keyword stuffing, while those with high compression ratios often feature noise, mixed-language content, or irregular formatting. The intermediate range consistently contains structured, coherent, and informative text.

While these qualitative insights are compelling, we seek a more rigorous quantitative assessment. To do this, we propose examining how compression ratio distributions change across datasets known to vary widely in quality. Specifically, we ask: *Do higher-quality datasets systematically exclude documents at the extremes of compression ratio distributions?*

To quantitatively investigate this, we analyze compression ratios over a random sample of 1 million documents from each of four corpora: C4, FineWeb-Edu, DCLM, and Twitter. These datasets span a broad spectrum of curation strategies and observed downstream quality. C4 is a minimally filtered Common Crawl dataset (Dodge et al., 2021); FineWeb-Edu applies heuristic and classifier-based filtering (Penedo et al., 2024); DCLM undergoes model-based filtering(Li et al., 2024); and Twitter consists of informal, noisy, and unstructured web content(Mohammad et al., 2018; Barbieri et al., 2018). Figure 3 clearly illustrates that as perceived dataset quality improves, compression ratio distributions become narrower and more centralized. Lower-quality datasets like C4 have broader distributions with substantial tails, whereas higher-quality datasets such as DCLM exhibit tight, centralized distributions, effectively dropping documents with extreme compression ratios.

These correlational observations strongly motivate a causal exploration: whether actively filtering documents based on compression ratio can concretely enhance downstream language model performance. This causal hypothesis is explored further in Section 3.1.

## 3.1 ALIGNMENT WITH EXISTING QUALITY SIGNALS

Figure 4 provides compelling evidence that compression ratio aligns closely with decisions made by the FineWeb-EDU quality classifier. Documents accepted by this classifier predominantly occupy an

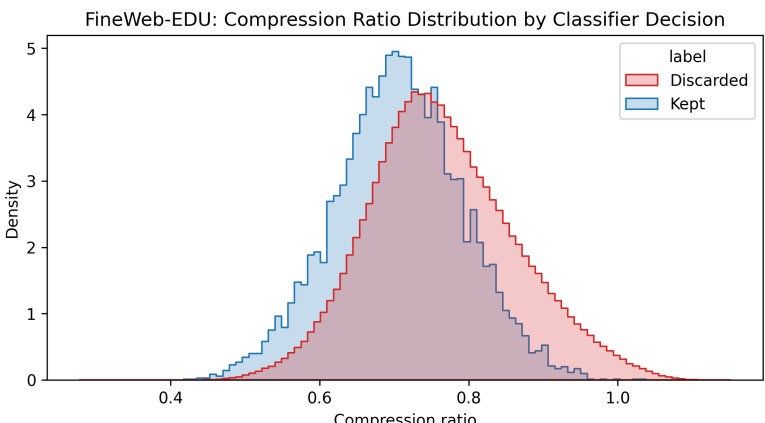

Figure 4: **Compression ratio mirrors many decisions made by the FineWeb–EDU quality classifier.** We plot kernel–density estimates of the LZ4 compression ratio for one million randomly-sampled documents that the classifier *keeps* (blue) versus *discards* (red). Lower ratios indicate text that is highly compressible (repetitive/boiler-plate), whereas very high ratios often correspond to noisy or degenerate content.

intermediate compression ratio range (approximately 0.65–0.80), reflecting text that is sufficiently informative yet neither overly repetitive nor excessively noisy. Conversely, documents discarded by the classifier display a broader compression ratio distribution, encompassing both highly repetitive (low compression) and noisy or irregular (high compression) extremes.

This alignment underscores compression ratio's utility as an intuitive, distributional proxy for textual quality, capturing much of the information implicitly leveraged by more computationally demanding, model-based classifiers. Nevertheless, the notable overlap between the "kept" and "discarded" distributions indicates inherent limitations of rigid thresholding strategies. Many documents near the decision boundaries remain challenging to classify accurately via compression ratio alone.

## 4    COMPEL: A METHOD FOR IMPROVING PRETRAINING DATA QUALITY

### 4.1    DESIDERATA

An effective data selection method for large-scale language model pretraining should satisfy the following criteria:

- **Performance**: It should improve downstream accuracy or generalization relative to unfiltered or naively filtered data.
- **Scalability**: It must operate efficiently over trillions of tokens.
- **Simplicity**: The method should be easy to implement, interpret, and integrate into existing preprocessing pipelines.

Compel is designed with these criteria in mind. It is an embedding-free, compression-based filtering method that improves data quality using a single, interpretable scalar: compression ratio. It requires no training, inference, or human labels.

### 4.2    BACKGROUND: COMPRESSION ALGORITHMS

Lossless compression algorithms reduce file size by exploiting redundancy in the input. In this work, we utilize LZ4 a fast, block-based compressor that employs LZ77-style sliding-window encoding. LZ4 is streaming-compatible and operates over raw byte strings, making it ideal for large-scale filtering pipelines. Unlike perplexity scores or embedding distances, compression operates without tokenization, external supervision, or model inference.

## 4.3 Filtering Criteria

Based on empirical observations from Section 3, we define a quality band of compression ratios: $\tau_{\min} = 0.65$ and $\tau_{\max} = 0.80$. Let $\mathcal{D} = \{x_1, x_2, \ldots, x_N\}$ be a collection of documents in a pretraining corpus. Documents with $\text{CR}(x) \in [\tau_{\min}, \tau_{\max}]$ are retained. Those falling outside this range are discarded. These threshold values were selected based on the inter-quartile ranges observed in high-quality datasets like DCLM and FineWeb-EDU, where well-structured, information-dense documents tend to fall within this band. This provides a simple, domain-agnostic filter that effectively removes both overly redundant and noisy examples.

## 4.4 Efficiency

`Compel` is highly efficient and trivially parallelizable. On a machine with 500 CPUs, the filter processes over 40,000 documents per second using LZ4 compression—an order of magnitude faster than typical perplexity- or classifier-based quality filters.

## 5 Experiments

### 5.1 Pretraining Corpora

We evaluate `Compel` on three widely-used corpora for language model pretraining:

- **FineWeb** (Penedo et al., 2024): A 15-trillion-token web crawl filtered with domain heuristics and safety constraints.
- **FineWeb-EDU** (Penedo et al., 2024): A 1.3-trillion-token educational subset extracted using a synthetic LLM classifier trained on LLaMA 3 70B-Instruct annotations.
- **DCLM** (Li et al., 2024): A 4-trillion-token corpus derived from Common Crawl, curated to support high-accuracy model training with limited compute.

For each corpus, we construct a *Compel-filtered* variant by removing documents whose LZ4 compression ratio falls outside the calibrated range:

$$\tau_{\min} = 0.65, \quad \tau_{\max} = 0.80.$$

Compression is applied to raw byte-level representations using a single pass over the dataset. Filtering is performed prior to tokenization and shuffling.

### 5.2 Model Configurations

For each Compel-filtered variant, we train two decoder-only Transformer models based on the LLaMA Touvron et al. (2023) architecture with rotary position embeddings and grouped-query attention. The **1.4B model** has 16 layers, a hidden size of 2048, intermediate size of 7168, 16 attention heads, and 8 key/value heads. The **8B model** consists of 32 layers, a hidden size of 4096, intermediate size of 14336, 32 attention heads, and 8 key/value heads. Both models are configured with a maximum sequence length of 4096. These values were selected based on standard configurations shown to perform well at this scale in prior open-source models. The 8B model follows the LLaMA3 hyperparameters (Touvron et al., 2023), which are tuned for stability and efficiency at high model capacity. All models are trained from scratch without weight reuse or initialization from pretrained checkpoints.

### 5.3 Training Protocol

The **1.4B model** is trained for 10,000 steps using a batch size of 1024 and sequence length of 4096, totaling 42 billion tokens. The **8B model** is trained for 40,000 steps under the same sequence length and batch size, totaling 167 billion tokens. Training is performed on TPU v4-128 pods.

We use AdamW optimization for all models. The 1.4B model uses a learning rate of $3 \times 10^{-4}$ with weight decay of 0.1. The 8B model follows the LLaMA 3 hyperparameters: a learning rate of $2 \times 10^{-3}$, weight decay 0.05, and 5000 warmup steps. All models are trained with cosine decay schedules and dropout rate of 0.1

## 5.4 Evaluation Suite

We evaluate each model using the standardized `lm-eval-harness` framework(Biderman et al., 2024), following zero-shot protocols on a diverse set of 13 natural language processing benchmarks:

- **Commonsense and Reasoning:** COPA(Roemmele et al., 2011), CommonsenseQA(Talmor et al., 2018), PIQA(Bisk et al., 2020), ARC(Easy/Challenge)(Clark et al., 2018), AGIEval-LSAT-AR(Zhong et al., 2023)
- **Reading Comprehension:** BoolQ(Clark et al., 2019), OpenBookQA(Mihaylov et al., 2018), WinoGrande(Sakaguchi et al., 2021), WSC273(Levesque et al., 2012), LAMBADA(Paperno et al., 2016)
- **Multichoice QA:** HellaSwag(0-shot, 10-shot)(Zellers et al., 2019)

**Metrics.** We report task-standard metrics for each dataset, primarily multiple-choice question-answering accuracy. We aggregate results using unweighted macro-average accuracy over all tasks.

## 6 RESULTS

We evaluate `Compel` across three widely-used, large-scale pretraining datasets—FineWeb, FineWeb-EDU, and DCLM—using two LLaMA-style model scales: 1.4 billion (1.4B) and 8 billion (8B) parameters. We measure model performance using macro-average accuracy across a suite of 13 diverse downstream benchmarks (see Section 5.4). The detailed benchmark results are summarized in Table 1, with key findings discussed below.

### 6.1 Impact of Compression-Based Filtering at 1.4B Scale

Table 1: **Results for 1.4B-parameter models across datasets.** `Compel` improves on 9 out of 13 benchmarks for FineWeb, 9 out of 13 for FineWeb-EDU and 11 out of 13 benchmarks for DCLM

| Benchmark | FineWeb | FineWeb + Compel | FineWeb-EDU | FineWeb-EDU + Compel | DCLM | DCLM + Compel |
|---|---|---|---|---|---|---|
| WSC273 | 0.571 | **0.604** | 0.571 | **0.550** | 0.630 | **0.637** |
| Winogrande | 0.528 | **0.534** | 0.532 | **0.525** | 0.528 | **0.539** |
| PIQA | 0.708 | **0.712** | 0.687 | **0.702** | 0.705 | **0.710** |
| OpenBookQA | 0.196 | **0.192** | 0.244 | **0.258** | 0.222 | **0.230** |
| LAMBADA | 0.369 | **0.379** | 0.337 | **0.339** | 0.499 | **0.501** |
| HellaSwag (10-shot) | 0.362 | **0.365** | 0.365 | **0.364** | 0.371 | **0.378** |
| HellaSwag (0-shot) | 0.365 | **0.369** | 0.378 | **0.384** | 0.375 | **0.379** |
| COPA | 0.630 | **0.730** | 0.690 | **0.730** | 0.690 | **0.710** |
| CommonsenseQA | 0.201 | **0.188** | 0.216 | **0.197** | 0.199 | **0.207** |
| BoolQ | 0.591 | **0.593** | 0.571 | **0.606** | 0.507 | **0.462** |
| ARC-Easy | 0.537 | **0.536** | 0.623 | **0.634** | 0.609 | **0.614** |
| ARC-Challenge | 0.230 | **0.218** | 0.282 | **0.296** | 0.276 | **0.265** |
| AGIEval LSAT-AR | 0.230 | **0.239** | 0.248 | **0.257** | 0.217 | **0.222** |
| **Macro avg.** | 0.424 | **0.435** | 0.442 | **0.449** | 0.466 | **0.470** |

**FineWeb** As shown in Table 1, `Compel`-filtered models improve accuracy on 9 out of 13 tasks compared to the heavily-filtered FineWeb baseline, yielding a notable macro-average accuracy increase of 1.1 points (from 42.4% to 43.5%). The largest improvements occur in commonsense reasoning benchmarks, notably COPA (+10%) and WSC273 (+3.3%), suggesting that removing documents outside the calibrated compression ratio range effectively eliminates subtle forms of noise, enhancing models' reasoning capabilities.

**FineWeb-EDU** Even though FineWeb-EDU is already filtered via strong classifier-based heuristics, COMPEL further enhances model performance. As Table 1 indicates, COMPEL filtering leads to improved accuracy on 9 of 13 tasks, resulting in a 0.7-point increase in macro-average accuracy (from 44.2% to 44.9%). Notable gains are observed on COPA (+4%) and BoolQ (+3.5%).

**DCLM** DCLM represents the current gold standard in dataset quality. Nevertheless, Table 1 reveals that `Compel` still offers improvements on 11 of 13 tasks, yielding a 0.4-point increase in macro-average accuracy (from 46.6% to 47.0%).

## 6.2 EFFECTIVENESS OF COMPEL AT LARGER MODEL SCALE (8B)

Our 8B experiments demonstrate that Compel's effectiveness persists at scale: it improves both macro and micro average accuracy despite the increased capacity and learning dynamics of larger models. This suggests that Compel's benefits are not confined to small or medium-scale settings. Instead, compression-based filtering enhances training efficiency even when models have greater capacity to absorb noise, indicating that the quality improvements it introduces are fundamental rather than compensatory. As model sizes continue to grow, such lightweight filters can play an increasingly important role in optimizing compute budgets while preserving or improving performance.

Task-level results reveal notable improvements in tasks such as BoolQ (+5.5%) and CommonsenseQA (+2.6%), despite minor regressions on ARC-Challenge (–2.4%) and WSC273 (–1.1%). These results highlight that Compel filtering scales effectively with increased model size, preserving or enhancing model capabilities and improving data efficiency without added computational complexity.

Table 2: **Results on 8B models trained on FineWeb.** COMPEL preserves downstream accuracy while improving perplexity.

| Benchmark | FineWeb | FineWeb + Compel |
|---|---|---|
| WSC273 | 0.809 | **0.798** |
| Winogrande | 0.682 | **0.679** |
| PIQA | 0.793 | **0.791** |
| OpenBookQA | 0.310 | 0.310 |
| LAMBADA | 0.638 | **0.648** |
| HellaSwag (10-shot) | 0.556 | **0.561** |
| HellaSwag (0-shot) | 0.562 | 0.562 |
| COPA | 0.860 | **0.850** |
| CommonsenseQA | 0.189 | **0.215** |
| BoolQ | 0.683 | **0.738** |
| ARC-Easy | 0.705 | **0.732** |
| ARC-Challenge | 0.396 | **0.372** |
| AGIEval LSAT-AR | 0.204 | **0.213** |
| **Macro avg.** | 0.570 | **0.572** |
| **Micro avg.** | 0.587 | **0.593** |

## 7 COMPEL QUALITATIVELY SELECTS HIGHER QUALITY TEXT

To better understand the kinds of documents filtered by compression ratio, we visualize examples from each compression band: low (<0.65), Compel-kept (0.65–0.80), and high (>0.80). We display the beginning lines of real web documents from each region, as shown in Figure 5.

The low-compression examples are dominated by boilerplate keyword stuffing and templated text (e.g., location-specific service listings repeated verbatim). High-compression examples tend to be noisy, short, or include artifacts like mixed-language metadata or hexadecimal tables. In contrast, documents retained by Compel exhibit higher information density and structural coherence—typically consisting of fluent, formal descriptions, educational explanations, or open-source project write-ups.

Qualitatively, Compel effectively discards content at both ends of the redundancy–entropy spectrum, keeping the center band where well-edited, human-authored text lies. This confirms the value of compression ratio as a simple yet powerful signal for improving pretraining data quality.

## 8 DISCUSSION AND LIMITATIONS

Compel was designed to address a fundamental challenge in large-scale language model training: identifying and filtering high-quality data from internet-scale corpora without incurring the cost of large models, supervision, or inference. While individual gains from Compel filtering are modest (+0.5–1.1 points), our results demonstrate that these gains are consistent across datasets and model sizes, achieved with negligible computational overhead.

| **CR** $< 0.65$ | $0.65 \leq$ **CR** $\leq 0.80$ | **CR** $> 0.80$ |
|---|---|---|
| Asus Laptop Virus Removal Asus Repair Service Area: Asus Laptop Virus Removal Toronto, Laptop Repair GTA, Laptop Repair Thornhill, Laptop Repair Richmond Hill, Laptop Repair Markham, Laptop Repair Vaughan, Laptop Repair Toronto, Laptop Repair Scarborough, Laptop Repair Newmarket, Laptop Repair Mississauga... | This project is solving the Asteroid Watchers challenge. AROs are essentially created by combining a telescope with a smartphone. If the telescope has drive motors controllable via a standard // protocol, it is commanded by the SkyWatch app on the phone. Telescopes without motors need an Arduino-driven stepper setup. We plan to publish a... | 根香港法律,不得在程中,向未成年人售或供令人醺醉的酒。 Under the law of Hong Kong, intoxicating liquor must not be sold or supplied to a minor in the course of business. Enjoy FREE DELIVERY for orders HK$1500 (net) or more, otherwise a delivery charge may apply. Hotline: +852 9063 3222 visa visa verified master master secure paypal... |

Figure 5: Beginning characters of representative examples excluded for low compression ratio (left), kept by Compel (middle), and excluded for high compression ratio (right). Compel retains the informative mid-range while filtering boilerplate or noisy text.

Prior work has shown that small, reproducible accuracy improvements can have outsize benefits at scale (et al., 2023; 2024a; Tirumala et al., 2023). For example, a 1-point improvement in accuracy at trillion-token scale can eliminate the need for hundreds of thousands of additional training steps or significantly reduce reliance on larger, more costly models.

By offering a simple, domain-agnostic filter that integrates seamlessly into existing pipelines, Compel provides a practical path toward higher-quality data at minimal cost—helping bridge the gap between dataset scale and dataset quality.

Our approach has several limitations. First, compression thresholds were manually tuned based on observed distributions from reference corpora, which may not generalize optimally across diverse datasets or content domains. While these thresholds proved broadly effective, our analysis did not systematically explore their sensitivity or optimality due to computational constraints. Future work could automate this process by using adaptive threshold selection techniques that optimize filtering performance on small validation sets or leverage unsupervised clustering over compression distributions.

Second, applying a fixed, global compression threshold across entire corpora inherently ignores variability across different content types or domains. More granular, adaptive filtering—especially tailored to specific domains such as code repositories, news, or social media content—could yield further performance gains.

Third, we positioned Compel as a final refinement step in existing filtering pipelines. Although this demonstrated its complementary nature and consistent improvements, we believe compression-based filtering may provide greater value when applied earlier in data pipelines. Early-stage filtering can reduce the amount of data subject to more expensive processing, making full pipelines more efficient.

Finally, the inherent simplicity of compression ratio as a scalar metric means it cannot capture semantic or nuanced linguistic content distinctions. Hybrid approaches that combine compression-based signals with lightweight semantic or lexical features could address this shortcoming, refining dataset quality further without incurring significant computational overhead.

Together, these limitations point to several promising directions: (1) adaptive thresholding based on unsupervised signal analysis, (2) domain-specific calibration, (3) early-stage integration into multi-pass pipelines, and (4) hybrid filters that combine compression with complementary lightweight signals.

## 9 CONCLUSION

Compel introduces a fast, model-free filtering signal that improves data quality across diverse corpora and model scales. As pretraining datasets continue to grow, compression-based filtering offers a practical path toward scalable, efficient, and interpretable data selection.

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

## A  APPENDIX / SUPPLEMENTAL MATERIAL

Optionally include supplemental material (complete proofs, additional experiments and plots) in appendix. All such materials **SHOULD be included in the main submission.**

