# OpenReview forum: "Curating High Quality Pretraining Data for Language Models via Compression Ratios"
_ICLR.cc/2026/Conference — Submitted to ICLR 2026_

### Official Review · Reviewer_BatL · 2025-10-21

**Soundness:** 2
**Presentation:** 2
**Contribution:** 2
**Rating:** 4
**Confidence:** 3

**Summary:**

The paper introduces Compel, a simple and fast method for improving the quality of pretraining data for language models. Instead of using complex classifiers or models, Compel uses a document’s compression ratio as a signal for text quality. The main idea is that well-written and information-rich documents fall in a “sweet spot” of compression and it's not too repetitive and not too noisy. By keeping only the documents within a certain range, this method can improve dataset quality efficiently. They test approach on FineWeb, FineWeb-EDU, and DCLM, and show small but consistent performance improvements on multiple benchmarks.

**Strengths:**

1. Using compression ratio as a filtering signal is a clever and refreshing idea. It’s not common in data filtering pipelines, and it avoids the need for training additional models or using expensive heuristics.
2. The experiments are solid and cover multiple datasets and model sizes. Results are consistent and clearly show that the method works.
3. The paper is easy to follow, with clear figures and explanations that make the concept intuitive even for readers without a deep technical background in data filtering.

**Weaknesses:**

1. The thresholds are manually set and may not work equally well for all domains or languages. It would be better if the paper show how sensitive results are to these values or suggested an adaptive method.
2. There’s limited theoretical justification for why this compression range works best or how it might relate to linguistic quality in different contexts.
3. The discussion of limitations could include more practical guidance , such as how users could combine Compel with other filters in real pipelines.

**Questions:**

1. How stable are the results if different compression algorithms are used instead of LZ4?
2. Does Compel work equally well for multilingual data or code datasets?
3. Since compression ratio is language-agnostic, is there a risk of keeping data with good structure but misleading or low factual value?
4. Could you share more details about the computational savings compared to perplexity-based filters?

---

### Official Review · Reviewer_QeRZ · 2025-10-29

**Soundness:** 3
**Presentation:** 3
**Contribution:** 3
**Rating:** 6
**Confidence:** 4

**Summary:**

This paper introduces a lightweight model-free filtering method that uses document compression ratio as a proxy for information density. By discarding documents with very low (repetitive) or very high (noisy) compression ratios, it improves the quality of pretraining data with minimal compute cost. Experiments on FineWeb, FineWeb-EDU, and DCLM show consistent gains across 13 benchmarks.

**Strengths:**

- Simple and interpretable idea that aligns well with intuition about data redundancy and noise.
- Requires no model training, labeling, or tokenization and is extremely efficient.
- Strong empirical evidence with diverse benchmarks.
- Qualitative examples nicely illustrate the filtering effect.

**Weaknesses:**

- Thresholds for compression ratio appear manually tuned and not rigorously justified.
- Missing analysis of compressor choice or sensitivity across domains.
- Gains are relatively small and not clear how meaningful.
- Focuses only on English web data; unclear generality to other languages.
- Limited exploration of combining compression with other lightweight signals.

**Questions:**

- How sensitive are results to the selected range?
- Would other compressors yield similar patterns?
- Have you tried domain- or language-specific thresholding?
- Does Compel meaningfully alter topic or style distributions in the data?

---

### Official Review · Reviewer_ZcWT · 2025-10-31

**Soundness:** 2
**Presentation:** 3
**Contribution:** 2
**Rating:** 2
**Confidence:** 4

**Summary:**

The paper presents a novel way of filtering large scale text data used for LLM pretraining. The novel method relies on the insight that compression ratio can be indicative of text quality, since it can detect repeated text (highly compressable) or malformed text (not compressable). This method does not require training models, it does not rely on underlying model embeddings, nor does it rely on human labels as standard classifier approaches do. Authors show that marginal gains can be achieved if their method is applied on top of existing filtered corpora such as fineweb and DCLM.

**Strengths:**

- They decide to invest their compute budget on more tests on datasets and LM instead of more compression methods for LM training. Makes the contribution stronger on downstream evaluations.
- The premise that a lightweight compression algorithm can be used to filter data to improve the performance of an LM is interesting and compelling.
- Experiments are conducted at good scales, at 1B and 8B scales.
- Their approach requires no human labels, which is an interesting property.

**Weaknesses:**

- The difference between C4 and DCLM in Figure3 does not seem so significant, both have similar top range in y axis, with similar spread on the x axis, differences do not seem to be so evident that compression captures the high noise of C4 and rewards the quality of DCLM.
- Figure 4 is used to convey that the compression ratio is well aligned with the FineWeb-EDU decisions, however in this plot the overlap between kept and discarded is large. It is not convincing that the compression ratio is well aligned with the edu classifiers. In fact, it seems to imply that if you discard, then it is of high compression, but the inverse causality (the relevant one) is not shown. If you use this CR metric, you will not get the edu classifiers (as also shown in Table 1 by the lower performance).
- In Figure 4, it also shows that examples with small compression ratios should be kept, which goes against your claim.
- There is no rigorous way of getting the decision thresholds for CR cutoffs, it is based on plot observations against Fineweb-EDU, which is not a scalable and well grounded approach. Additionally, at 0.65 there are more documents to be kept than discarded, it is not clear why that would be the lower bound cutoff…
- For Figure 4, they should have tried with more compression algorithms besides LZ4 to better strengthen their choice, this is not an expensive plot to run as it is CPU-only and can be done on a subset of the data. LZ4 does not have entropy coding as most compression algorithms do, which makes it fast, but it might not be the best choice.
- FineWeb and DCLM are built from raw, crawled text that is extremely hard to filter. Compel is only applied on top of these datasets, not on raw text, so it does not show to be a replacement for such approaches, but potentially as a way to slightly improve over already highly curated datasets. If compared against the edu method, Table 1 shows that the increase from Fineweb to EDU (0.018) is higher than from fineweb to fineweb+compel (0.011), which if the results in the table are significant, is a large difference.
- Compel claims to filter without incurring the costs of models, supervision or inference, however all results are over datasets that were already filtered in that way, so compel can only be applied after those costly steps have been done, it is not proved its efficacy in raw corpora (over which people applied models, etc… as claimed by the authors).
- The improvement over FineWeb-EDU of 0.007 and over DCLM of 0.004 seem small, given that there is no std of runs or statistical significance tests, it is hard to understand if these are meaningful, but seem small compared to the improvements over FineWeb.
- Table 1 does not say on how many tokens the models were trained. It is hard to know if they were trained well or if they are undertrained.
- The claims are that the modest gains they achieve, if done at larger scale, can have a very large impact, however this is never proven. I understand there are compute limitations, but these claims cannot be made if they cannot be proven.
- I notice they mention some of my concerns in section 8, which means the authors are aware of such limitations. I think the paper is not finished. Work towards those directions is needed to have something that is concrete, what is presented is a nice proof-of-concept but not a finished paper.
- They show only 1 baseline of filtering which is the EDU baseline, that outperforms their approach, more baselines or comparisons are needed to strengthen their claim. There are many works on filtering data, and they perform no comparisons.
- Claims on efficiency are made, but not proven/shown.

**Questions:**

- If DCLM performs better, why is the work trying to replicate the choices made by FineWeb-EDU?
- In Table 2 it is not clear how the macro and micro average were done as it is not in the main text. Could this be explained?
- Could they provide, in the appendix, a more thorough study/comparison of the high compression ratio text? They say it is artificial text, but the example provided simply mixes two languages which is easily discardable by language id or any other easy metrics. What does English-only artificial text that gets filtered out look like? The examples in section 7 are the same ones as in the beginning, a more thorough analysis is needed to improve the quality of the paper.
- They claim their approach is more lightweight and efficient. There are, however, no speed, or compute costs comparison. Is it possible to run baselines (such as the edu classifiers), and compare the time spent on all approaches, and how much does it compare to the training time of the LLMs.
- For large scale LLM training, is data filtering a bottleneck in terms of time? Is it a relevant percentage compared to time of training and development of the models?
- I want to see how compression scales with quality of training, i.e. compression rate vs LLM performance, this is not done. I understand that compute is limited, but at least 1 experiment should be done, the scale does not need to be 1B, it can be at lower scale (e.g. 500M scale, or even lower at 200M scale and evaluated only on tasks such as HellaSwag and not with QA)
- Could the authors show systematically in texts they know are information dense and not, that compressing them indeed reveals the property they claim that CR can effectively correlate with this. I do not mean over a large dataset as they did. I mean over a small set of examples that are dense and not dense and see how well the compression predicts the density.
- How was the lower cutoff value of 0.65 chosen?
- Why was MMLU not included in the tasks?

---

### Official Review · Reviewer_G5rH · 2025-11-02

**Soundness:** 3
**Presentation:** 3
**Contribution:** 2
**Rating:** 4
**Confidence:** 4

**Summary:**

This paper proposes a compression-based data filtering method, Compel, for language model pretraining. This method monitors the compression ratio of documents and selects documents with a compression ratio within a range, where they calibrate the threshold using high-quality reference datasets. The paper shows that their method improved downstream performance across 3 large datasets based on evaluation across 13 tasks.

**Strengths:**

1. The proposed method is simple, clean, and most importantly, very scalable. It can be a high value for pre-training data curation.

2. Experiments and their setup are very clearly presented and easy to follow.

**Weaknesses:**

1. Compel needs to be compared with the baseline classifier- and perplexity-based filtering methods for both performance and computational cost dimensions. Showing a conservative improvement over the base datasets is not enough to judge the benefits of the method. For instance, since a high-quality dataset needs to be used for rthe eference threshold range, the same dataset can be used for classifier-training to compare.

2. It is quite understandable that the experimental setup is relatively small (in terms of token budget) due to the cost of pretraining. However, for a pre-training data filtering work, it is important to understand how robust the method is when the token budget is larger, since the language models are commonly pretrained with trillions of tokens. Additional ablations that have an increasing token budget (to at least show the trend) are required.

3. Figure 4 could be misleading -- documents accepted occupy the selected compression range; however, there is no distinct trend showing that discarded documents are out of this range.

**Questions:**

Why did you not use Fineweb-Edu as a comparison point rather than a pre-training dataset? The results from 1.4B show higher performance with Fineweb-EDU (0.442 vs 0.435)

I would like to see how Compel Fineweb filtering compares with Fineweb-Edu itself (LLM classifier), also in the 8B model.

---

### Meta-Review · Area_Chair_uHJA · 2025-12-15

**Summary:**

Initial scores were 6, 4, 4, 2 (average: 4.0). Reviewers acknowledged the simple, scalable approach but raised concerns about: (1) lack of comparison with classifier- and perplexity-based filtering baselines, (2) manually-tuned thresholds without rigorous justification, (3) marginal improvements (0.004-0.011) without statistical significance testing, (4) method only tested on already-filtered datasets (FineWeb, DCLM) rather than raw corpora, and (5) missing efficiency/cost comparisons despite claims of being "lightweight."

**Reviewer Concerns:**

**No author response was provided.** All reviewer concerns remain unaddressed.

**Reviewer Scores:**

No response provided. All reviewers would most likely maintain their scores.

- **Reviewer G5rH (initial: 4):** Core concerns about missing baseline comparisons and token budget scaling are fundamental methodological issues requiring new experiments.

- **Reviewer ZcWT (initial: 2):** Extensive list of concerns indicates the reviewer views this as incomplete work ("not a finished paper").

- **Reviewer QeRZ (initial: 6):** Most positive assessment values the simplicity and efficiency, but concerns about threshold tuning and small gains persist.

- **Reviewer BatL (initial: 4):** Concerns about manual threshold setting and lack of theoretical justification remain without author clarification.

---

### Decision · Program_Chairs · 2026-01-26

Reject